# A General Prescription for Semi-Classical Holography

Budhaditya BHATTACHARJEE[a*] & Chethan KRISHNAN[a†]

$^a$ Center for High Energy Physics,
Indian Institute of Science, Bangalore 560012, India

## Abstract

We present a version of holographic correspondence where bulk solutions with sources localized on the holographic screen are the key objects of interest, and not bulk solutions defined by their boundary values on the screen. We can use this to calculate semi-classical holographic correlators in fairly general spacetimes, including flat space with timelike screens. We find that our approach reduces to the standard Dirichlet-like approach, when restricted to the boundary of AdS. But in more general settings, the analytic continuation of the Dirichlet Green function does not lead to a Feynman propagator in the bulk. Our prescription avoids this problem. Furthermore, in Lorentzian signature we find an additional homogeneous mode. This is a natural proxy for the AdS normalizable mode and allows us to do bulk reconstruction. We also find that the extrapolate and differential dictionaries match. Perturbatively adding bulk interactions to these discussions is straightforward. We conclude by elevating some of these ideas into a general philosophy about mechanics and field theory. We argue that localizing sources on suitable submanifolds can be an instructive alternative formalism to treating these submanifolds as boundaries.

---
*budhaditya95@gmail.com
†chethan.krishnan@gmail.com

# 1 Introduction

Immediately after the discovery of the AdS/CFT duality [1], a semi-classical holographic correspondence between the bulk and the boundary of AdS was presented in eg., [2, 3, 4, 5, 6, 7]. These papers use perturbative bulk physics to determine correlators and states on the boundary. One of the rudimentary quasi-tests of the duality was that objects calculated this way had the form expected in a conformal field theory (CFT). A bit later, it was noted by Hamilton, Kabat, Lifschytz and Lowe (HKLL) [8] that using the correspondence, one could also express local bulk physics (again at the semi-classical level) in terms of non-local operators in the boundary theory. This was the beginnings of the idea of bulk reconstruction.

One of the striking features of these developments is that none of them rely on the fine details of the dual theories. This is remarkable, and makes one wonder: how much of the semi-classical description of holography is actually tied to AdS? Is it possible to give a prescription for computing correlators on the holographic screen via bulk calculations in more general spacetimes (or regions of spacetime)? How about bulk reconstruction from appropriate screen data? Of course, one cannot expect the holographic correlators one computes in a general non-AdS setting to turn out to be those of a conformal field theory (or perhaps even a local theory). But that does not answer the question whether a semi-classical holographic description can be found at all (eg., do there exist a natural set of boundary/screen correlators that one can compute anti-holographically?), and if so, how best to formulate it.

In this note, we will observe that the general structure of the AdS/CFT correspondence uncovered in the early papers on the subject, has a very natural adaptation to more general settings. We will argue that the manner in which the semi-classical bulk physics of a region of spacetime is encoded on a holographic screen in terms of sources, condensates and correlators is essentially universal in large classes of spacetimes[1]. The general prescription we give will be in terms of bulk sources (localized at the screen) and homogeneous modes, instead of the usual normalizable and non-normalizable modes familiar from AdS [4, 5]. Instead of working with boundary values of bulk fields, our Euclidean correspondence will be built on bulk sources localized on the holographic screen. This means that we will work *not* with Dirichlet Green function (which is defined by a vanishing condition at the screen) but with the standard Euclidean bulk-bulk Green function that dies out at infinity. In particular, this means that when analytically continued to Lorentzian signature, this will lead us to the

---

[1] We will describe the classes of spacetimes in which we expect our claims to hold, somewhat imprecisely, in the next sections. Our claims depend on the existence/uniqueness of solutions of partial differential (wave) equations, which may have some subtleties (especially in Lorentzian signature) in some situations. A very concrete example of our prescription is provided in this and a companion paper [14] for the case of flat space with a $\mathbb{R} \times S^d$ screen, but we expect that the prescription holds somewhat more generally.

standard Feynman propagator as we would like, for causality reasons. In Lorentzian, we also find a bonus feature: the source and the Green function do not uniquely fix the solution, we also have the freedom to add a homogeneous mode that is regular everywhere. This is the analogue of the normalizable mode in AdS. From the on-screen value of the homogeneous mode, we will also be able to reconstruct the bulk field starting with the closely related spacelike Green function [8, 11] and extracting an HKLL-like kernel.

We will see that our prescription has a simple map to the standard AdS/CFT correspondence, when restricted to AdS. To understand this connection it is useful first to note that sources placed on the holographic screen will depend on the value of the holographic coordinate (let us call it $r$) of the screen ($r = R$), and therefore will lead to correlators that depend on this $R$. A crucial point is that the $R$-dependence of a general bulk solution is (in general) dependent on the angular harmonics[2]. The beautiful fact about AdS is that the angular harmonics dependence of $R$ dies out near the boundary, and in fact the $R$-dependence factorizes out. This enables us to define a new class of correlators very simply, where this $R$-dependence can be compensated. These $R$-independent correlators are precisely those of a CFT. The deep reason why this happens in AdS is of course because the bulk isometries act as the conformal group on the asymptotic boundary. In practice this means that one can replace the bulk-to-boundary propagator that one typically uses in AdS (basically a type of Dirichlet Green function) with the boundary limit of the bulk-to-bulk propagator (the Euclidean Green function that dies out at infinity) as long as one keeps track of some simple $R$-dependent scalings. Our on-screen-source prescription, together with this rescaling can be shown to be equivalent to the standard Dirichlet-like[3] AdS/CFT prescription. In other words, these two prescriptions are equivalent in AdS. Our claim is that when trying to move away from AdS however, the source prescription is more natural.

In this note, we will focus on laying out the general statements. We will also present some results for an example of some interest: 3+1 dimensional Minkowski space, with the holographic screen chosen to be a spherical box $\mathbb{R} \times S^2$ of finite radius $R$. A more detailed analysis of this example will be left for a future paper [14]. It is possible to explore this example in some detail and calculate holographic correlators, Witten diagrams and HKLL-like smearing functions in flat space.

Let us summarize. When trying to formulate holographic correspondence in general spacetime regions, a useful object is the general solution of the bulk equations of motion with sources placed on the holographic screen. Unlike in AdS, the notion of normalizability

---

[2]Note that the general solution is a sum of products of the angular and radial parts: very schematically, $\sum_l R_l(r) Y_l(\Omega)$. This means that typically one cannot factorize out the $r$-dependence.

[3]We say Dirichlet-like, instead of Dirichlet, because the boundary value of the field is only fixed up to this overall scaling in the standard AdS/CFT correspondence.

vs non-normalizability of the solutions of the bulk field equations becomes awkward in general settings. But a natural generalization presents itself in many cases: the homogeneous and inhomogeneous pieces in the solutions of the bulk field equations with an arbitrary source at the holographic screen. This structure is general enough to allow the entire semi-classical holographic correspondence to go through in fairly general classes of spacetime regions including flat space, and it reduces in a suitable sense to the usual prescription in the AdS case. It is also immediate to see that even though our statements are phrased in terms of free theories, perturbatively adding interactions is as straightforward as it is in AdS. We will also demonstrate that the extrapolate and differential dictionaries match, as they do in standard AdS/CFT.

We elaborate on these ideas in the next sections. One point of view that emerges from these discussions is that it might be worthwhile on general grounds, to consider sources localized on submanifolds as a general approach to formulating dynamics. This could be of some interest even beyond holography. It serves as as an alternative to the usual formalism familiar from mechanics and field theory, where we describe dynamics via boundary and/or initial value data on submanifolds, together with boundary terms and the like. Instead, here the idea is to consider boundary conditions that always die down at (possibly Euclidean) infinity, but allowing sources localized on appropriate submanifolds as a mechanism for capturing the physics.

One final note on terminology before we proceed: we have tried to be consistent in our use of the words screen and boundary in this paper, because they are logically distinct. We compute the holographic correlators on the screen, but the asymptotic boundary of the geometry often lies elsewhere. However, when there is unlikely to be any confusion, we have decided to sometimes *not* over-use neologisms – eg., what should more accurately be called the bulk-to-screen propagator, on occasion we call the bulk-to-boundary propagator.

## 2 Euclidean Holography: Sources at the Holographic Screen

Let us start with Euclidean signature. Consider a connected region of a general spacetime, with a codimension one hypersurface as its boundary/holographic screen[4]. Intuitively, we will think of the holographic screen as a hypersurface defined by two properties[5]: (a) it must separate the spacetime into two disconnected regions, (b) it should have a smooth deformation to the empty hypersurface that respects property (a). The first condition is obvious. The second condition means that the screen belongs to a one parameter family of

---

[4]We can relax these conditions (eg., multiple disconnected boundaries) by making various trade-offs. But we will stick with this for concreteness. Note also that 1+1 dimensions requires some special treatment.

[5]We assume that the spacetime is topologically trivial.

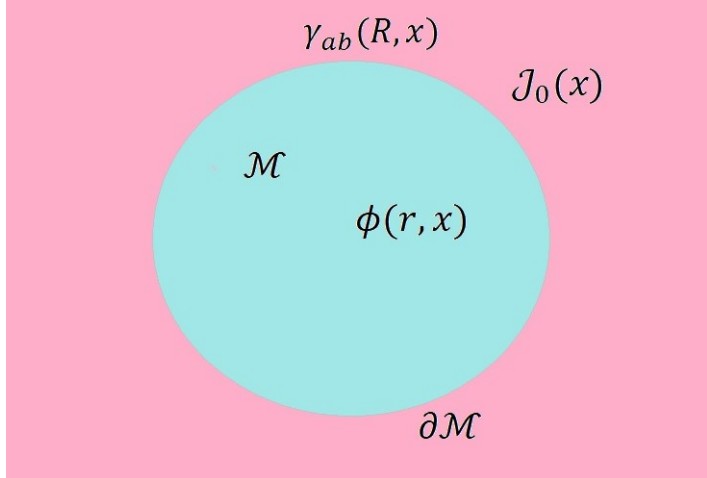

Figure 1: The blue region denotes the Euclidean $d+1$-dimensional bulk $\mathcal{M}$. The $d$-dimensional boundary aka holographic screen is $\partial\mathcal{M}$, the bulk field is $\phi(r,x)$, and the source on the screen is $J_0(R,x)$. The coordinate $r$ represents the bulk foliation and $r = R$ will be taken as the screen. The region $\mathcal{M}$ can be a sub-region of spacetime, and the $x$ is $d$-dimensional.

screens connected to the trivial (ie., non-existent) screen. Note that this parameter can be thought of as the holographic direction. The above conditions mean that (for example) in three dimensional Euclidean space, topological spheres and infinite cylinders are acceptable holographic screens in our book, but not infinite planes. This is consistent with the intuition that a holographic screen can be shrunk.

Using the $d+1$ coordinate freedoms available to us, we can work (without loss of generality) in a gauge where the bulk metric takes the form

$$ds^2 = dr^2 + \gamma_{ab}(r,x)\,dx^a dx^b. \tag{2.1}$$

We will take the holographic screen to be at $r = R$. While the form of the metric above is just a coordinate choice and does not lead to any serious loss of generality, in making the above choice of screen, we are being restrictive. We are taking our screen to be normal to the radial coordinate, but since we have already made use of all our freedoms to pick the form of the metric, this cannot be accomplished without some loss of generality. But we will make this choice for concreteness and convenience anyway, even though many of the statements we make below evidently go through even for more general screens[6]. We will take the metric $\gamma_{ab}$ to be Euclidean in this subsection, and Lorentzian in the next. In other words, the Lorentzian time direction is contained in the $a, b, ...$ directions.

---

[6]Let us emphasize the obvious here however: this choice does *not* mean that we are constraining ourselves to spherical symmetry.

We wish to holographically describe the dynamics of a bulk field $\phi(r, x)$ with the action $S_{bulk}$ around a semi-classical bulk background. We wish to do this as much as possible by analogy with AdS. In Euclidean signature, we will make the following guess for the proposal, and it will also serve as a stepping stone to the Lorentzian case, which involves some further subtleties. Without further ado, let us first state the *Euclidean prescription*:

- Find the general solution of the bulk wave equation with an arbitrary source $J_0(R, x)$ at the holographic screen. More precisely, we mean that we will consider solutions of the bulk free field equations of motion, with a bulk source $J_0(R, x)\delta(r - R)$ in spacetime:

$$(\Box - m^2)\phi(r, x) = J_0(R, x)\delta(r - R) \tag{2.2}$$

Here $\Box$ is the (Euclidean) Laplacian. In a non-singular Euclidean geometry, the solution will take the form

$$\phi(r, x) = \int_{\partial \mathcal{M}} d^d x' \sqrt{\gamma(R, x')} \, G_E(r, x; R, x') J_0(R, x') \tag{2.3}$$

where the Green function $G_E(r, x; R, x')$ is the bulk-to-bulk propagator (with one location taken to the screen). This object is defined as the solution to

$$(\Box_{r,x} - m^2) G_E(r, x; r', x') = \frac{\delta^d(x - x')\delta(r - r')}{\sqrt{\gamma(r', x')}}, \tag{2.4}$$

and it can be checked as usual from this that (2.3) solves (2.2). Note that in Euclidean signature, the wave equations are elliptic equations, and therefore we expect these solutions to exist, be regular (except perhaps at the source) and be unique[7]. Note also that in order to write the above form, we have assumed that the bulk fields are free. We will be able to perturbatively add interactions via generalizations of Witten diagrams etc., but truly strong coupling bulk effects are as difficult here, as they are in AdS.

A key point is that the choice of the Green function is not completely fixed without some further input, a boundary condition of some sort. We will take this condition to

---

[7]A source-less Euclidean "wave" equation is a Laplace-type equation, and by analogy with flat space, generically (ie., in generic dimension and for generic angular quantum number) we expect it to have two kinds of solutions. One that is singular at the origin, and another that is singular at infinity. A source at a finite radius will create a perturbation that should die down at infinity, and should therefore be expressible exclusively in terms of solutions singular at the origin. Linear combinations of singular solutions can result in a *finite* shift in the location of the singularity. This is basically the idea behind the familiar multi-pole expansion in electrostatics. When one goes over to Lorentzian signature, it turns out that the divergent solution at infinity ceases to exist, and gets replaced by solutions that are regular everywhere. In the next section, we will see that these regular solutions play a role analogous to AdS normalizable modes.

be that it should vanish at infinity, fixing it to be the conventional Euclidean Green function (we have incorporated this choice already in the notation with the subscript $E$). Our motivation for this choice is that we want the Green function to analytically continue to the bulk Feynman propagator when we analytically continue the time coordinate to Lorentzian signature. We will see in the next section that there exists physically interesting candidate $\gamma_{ab}$'s for which this can be explicitly seen.

- Compute the bulk on-shell (semi-classical) partition function, or (in practice) the classical action, of this solution. We will consider two prescriptions for doing this. One where we compute the action by integrating all the way to infinity, and the other where we only integrate up to the screen. In the limit where the radius goes to infinity (perhaps in some suitable scaling limit), these two prescriptions presumably do not make a difference in the screen correlators they compute, but at finite radius they will lead to different answers. We will motivate and discuss the first prescription (integrating to infinity) here, and relegate the second prescription to an appendix.

The motivation for integrating to infinity is three-fold. Firstly, our Euclidean Green function is defined to be vanishing at infinity. This suggests that our bulk actions should also be computed by integrating to infinity. Secondly, we wish to define the hologram of the entire spacetime, and not just of the region in its interior. This is related to our prejudice that we suspect holography to be screen-covariant in some suitable sense[8]. Thirdly, we will see that this prescription has the bonus feature that the so-called *extrapolate* and *differential* dictionaries for computing screen correlators, coincide[9]. This is a feature that is true in the usual AdS/CFT correspondence [15]. Its validity in the present setting, we will view as another indication that we are doing something right. Let us demonstrate this.

In the scalar two-derivative theory, the action for a solution of (2.2) will get contributions only from the two sides of the screen: this is because by an integration by parts, the bulk contributions vanish on-shell. These contributions are

$$S_{bulk}^{\pm} = \lim_{r \to R^{\pm}} \int_{\partial M} d^d x \, \sqrt{\gamma(r, x)} \, \phi(r, x) \partial_r \phi(r, x). \tag{2.5}$$

The total action is the sum of these two pieces, but with an appropriate negative sign that needs to be fixed according to the outward direction of the normal at $r = R$. Noting that $\phi$ is continuous across the screen, and using the integral of (2.2) across

---

[8]Of course, at special screen locations, like at the conformal boundary of AdS, the dual correlators might simplify.

[9]We thank Feroz Hatha and Ajay Mohan for discussions on this and Ashoke Sen for emphasizing to us that this is a good thing.

the delta function to relate the discontinuity in $\partial_r \phi$ to $J_0$, this leads to the total bulk action

$$S_{bulk} = \int_{\partial M} d^d x \sqrt{\gamma(R, x)} \, \phi(R, x) \, J_0(R, x). \tag{2.6}$$

Using (2.3), the $\phi$ in the above expression can immediately be written as a functional of $J_0(R, x)$. This lets us calculate correlation functions of operators $O(x)$ on the screen dual to the sources $J_0(R, x)$ via functional derivatives with respect to the $J_0(R, x)$, in analogy with the differential dictionary for correlator calculation in AdS. When evaluated at zero-source, this procedure leads to a vanishing 1-point function, and a two-point function of the form

$$\langle O(x')O(x'') \rangle = G(R, x'; R, x''). \tag{2.7}$$

In other words, the differential prescription yields an on-screen 2-point function that is just the restriction of the bulk 2-point function to the screen. In yet other words, what we have done here amounts to a demonstration of the equivalence between the differential and the extrapolate dictionaries! Note that this depended crucially on the fact that we are working with a source problem and not a boundary value problem. In particular, the Dirichlet Green function vanishes on the screen[10]. In fact, the reason why a Dirichlet-like problem in AdS leads to the matching between the extrapolate and differential dictionaries [15] is precisely because the problem there is only Dirichlet-*like*. The presence of the overall scaling (see (2.8)) in this Dirichlet-like problem enables us to re-interpret the solution in terms of a source problem, as we discuss later in this section.

Let us make some comments.

One might (or at least we did) worry that since we are working with sources instead of boundary values, we might run into trouble with these calculations because of potential divergences at the location of the sources. This is a false concern. A familiar example that clarifies this point is to consider the electrostatic potential due to a (uniformly) charged shell: the potential is finite at the shell. The key point is that to get a divergence at the source, we need the source to be localized in sufficiently many dimensions, eg., localized in all coordinate directions[11].

---

[10]There is a bit of a subtlety here, because it is the radial derivative of the Dirichlet Green function, and not the Green function itself, that propagates the boundary value of the field to its bulk value (see eg., [16]). Nonetheless, it is easy to convince oneself that there is no simple algebraic relation between the restriction of the Dirichlet Green function (or its derivative) and the on-screen Green function, in a general spacetime.

[11]Stated differently, remember the familiar fact that a uniform charge on an infinite plane leads to a constant electric field in the bulk. This is because as you step back from the plane, you "see" more of the plane and therefore more of the charge on it, so the field does not fall too fast. One cannot have a localized singularity in the field without the field dropping fast.

Another related possible confusion (to which we were again victims of) is that since the field is finite at the source, why don't we simply view this as a Dirichlet problem where the field is held fixed at the screen? The answer is that while the field may be finite, its value now depends on the Green function as well as the source, and so it makes a difference, what is the natural *a priori* data for the problem. That data is provided by the sources, and not the boundary values, in our prescription. If we want to work with Dirichlet data, we need to make sure that the Green function vanishes at the screen, which is what defines the Dirichlet Green function.

Let us also note that even though the field itself is not divergent, the screen correlators that we calculate will have singularities (as can be easily checked for specific examples), when two operators coincide. This is physical, and should be compared to the isomorphic phenomenon in Euclidean AdS/CFT. Once one goes over to Lorentzian, these turn into null separation singularities at the boundary/screen, which also applies in our case.

To summarize, we decree that the bulk (semi-classical) partition function in a spacetime region is defined for field configurations with sources at the screen, and is a functional of those sources. From the perspective of the dual theory on the screen, the same source also (apparently) couples to the operators dual to the bulk field. In AdS, that the boundary values of bulk fields are sources for the boundary theory is well-discussed, but it is also true (though less emphasized) that these boundary values can also be understood in terms of limits of sources for the *bulk* theory, at least for scalar fields. We will clarify this point momentarily, and make some comments about more general fields in the conclusions. In any event, what we have done here is to reverse the logic a bit and to treat localized bulk sources instead of boundary values as the key objects. This is a useful step in going to the Lorentzian signature when we are in a more general situation than AdS, as we will see.

**AdS:** In this subsection, we will clarify the connection between our prescription and the standard Euclidean AdS/CFT [12]. In [2, 3] one seeks solutions of the bulk wave equations that are regular in the interior. The resulting solutions are necessarily of the so-called non-normalizable variety, and at the boundary (defined by $z = 0$ in a standard choice of the Poincare patch coordinates) their behavior (for scalar fields) is of the form

$$\phi(z, x) \to z^{d-\Delta}\phi_0(x) + \cdots \tag{2.8}$$

where $\Delta$ is determined by the mass of the scalar [13]. Note the remarkable fact that the $z$-dependence has factorized out, which is a special feature of AdS – in a general spacetime, the $z$-dependence and the angular harmonic dependence will mix and the full solution can only be written as a sum of products, not a single product.

---

[12]See [9] for a discussion of the usual AdS/CFT correspondence that is adapted to our discussion here.

[13]We will assume here that $\Delta > d/2$ to avoid some technicalities. With a bit more nuance, we can extend the discussion all the way to the CFT unitarity bound, $\Delta = \frac{d-2}{2}$.

In any event, this $\phi_0(x)$ is interpreted as the source to which the boundary operator couples. The bulk solution can be written in terms of this $\phi_0(x)$ as

$$\phi(z, x) \sim \int d^d x' \frac{z^\Delta}{(z^2 + (x - x')^2)^\Delta} \phi_0(x') \tag{2.9}$$

$$\equiv \int d^d x' G_{EAdS}(z, x; x') \phi_0(x') \tag{2.10}$$

where the $G_{EAdS}$ is usually called the bulk-to-boundary propagator in standard AdS/CFT. Our prescription on the other hand is that we should work with the bulk-to-bulk propagator itself, but with a source $J_0$ that couples to it at the boundary. We place the source close to the boundary at $z' = \epsilon$ and the solution takes the form

$$\phi(z, x) = \int d^d x' \sqrt{\gamma(\epsilon, x)} \, G_\Delta(z, x; \epsilon, x') J_0(\epsilon, x') \tag{2.11}$$

$$\sim \int d^d x' \frac{z^\Delta}{(z^2 + (x - x')^2)^\Delta} \epsilon^{\Delta - d} J_0(\epsilon, x'). \tag{2.12}$$

The first line introduces the bulk-to-bulk Green function. In the second line we have used its explicit form that can be looked up from [10] and the fact that $\epsilon$ is small to write down its leading behavior in $\epsilon$. Altogether, this means that boundary correlators computed via functional derivatives with respect to $\phi_0(x)$ (like in standard AdS/CFT) and $J_0(\epsilon, x)$ (like we do) differ merely by a simple $\epsilon$ dependent factor that is trivially incorporated into the prescription. If we kept track of the precise numerical coefficients of the Green functions above, the precise map turns out to be

$$\phi_0(x) \leftrightarrow \frac{\epsilon^{\Delta - d}}{2\Delta - d} J_0(\epsilon, x). \tag{2.13}$$

Our prescription clarifies what makes AdS special: in AdS there is a specific scaling of the sources that results in correlators that are independent of the location of the screen, at least when the screen is close to the asymptotic boundary. This is a result of the conformal action of the bulk isometries on the AdS boundary. In particular, in a general spacetime/region where we do not expect conformal invariance, but nonetheless expect holography to hold in some suitable sense, it should be clear that we should *expect* scale dependence. Indeed, in flat space, the correlators we find are $R$-dependent[14].

The bottomline is that in AdS, two kinds of PDE problems both lead to the same kind of on-screen correlators. The first is the Dirichlet boundary value problem familiar from the

---

[14]It would of course be interesting to see if the correlators in flat space have some other transformation (Poincare? BMS?) which becomes manifest in the limit where $R$ goes to infinity. It may be interesting to appropriate scaling limits that reveal these symmetries.

standard AdS/CFT correspondence. The second is a source problem of the type (2.2) with a source localized on the screen, and a Green function that vanishes at infinity. What our discussion above demonstrates is that both these problems lead to the same on-screen correlators if the screen is at the conformal boundary of AdS, and therefore, standard AdS/CFT cannot really tell them apart. This is perhaps an interesting curiosity when one is dealing with AdS/CFT, but its real significance emerges when one is trying to move away from AdS: it is the source problem that generalizes nicely, when one is not in AdS. We will see further evidence for this when we consider Lorentzian holography.

## 3  Lorentzian Holography: Homogeneous Solutions

We will now consider Lorentzian spacetime regions of the form (2.1). We will take the (Lorentzian) time to be one of the $a$, $b$, ... directions. The paradigmatic example we will have in mind will be $d + 1$-dimensional flat space with an Einstein-static screen at $r = R$:

$$ds^2 = dr^2 + (-dt^2 + r^2 d\Omega_{d-1}^2) \tag{3.1}$$

Note that (Poincare) AdS also falls into the same structure as (2.1). We expect that the claims we make in this section will hold somewhat more generally than either of these examples. In particular, regions of spacetimes which are "flat enough" (see eg., figure 2) should satisfy them. At least in cases where a $d+1$-dimensional region is bounded by a timelike hypersurface, such that the intersection of this surface with a constant time slice (Cauchy slice) is topologically $S^{d-1}$, we expect that the claims of this section have a chance of holding[15]. See figure 2 for an example region of this type carved out in the exterior Schwarzschild geometry.

We now present the statement of the *Lorentzian Prescription* in the following form:

- Find the general solution of the bulk wave equation with an arbitrary source at the holographic screen. As can be checked explicitly for the special case of (3.1), this has

---

[15]This is just the statement that we are considering tube-like spacetime regions bounded by timelike holographic screens. One possible subtlety is that solutions of wave equations in such a tube-like region could leak out through the top/bottom, and this could qualitatively change the solution structure, eg., think about the Penrose diagram of de Sitter space. But at least if in the Penrose diagram, the spacetime region (together with its timelike screen) is represented by a closed region, we suspect that the region is sufficiently similar to flat space that the prescription we give will hold. Clearly, this should be viewed as a plausible *sufficient* condition, and it is satisfied by regions like the one in figure 2. But these conditions may not be *necessary*. AdS Penrose diagram does not have the same structure, but AdS is consistent with a version of our prescription anyway. It is clearly of interest to make a more precise statement about the class of spacetime regions for which our prescription holds, this will require statements about solutions spaces of hyperbolic partial differential wave equations in various geometries. In practice, we will keep (3.1) in the back of our minds in the following discussions.

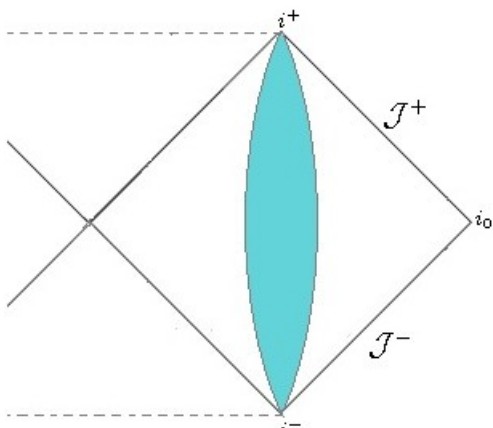

Figure 2: The blue region is bounded by a timelike tubular holographic screen in the exterior Schwarzschild geometry, and we expect our prescription to have analogues there. Note that this Penrose diagram stands for a slice of the geometry: each point is to be treated as a point, and not as a sphere, so that a slice of the "tube" can be meaningfully depicted.

the form

$$\phi(r, x) = \int_{\partial \mathcal{M}} d^d x' \sqrt{-\gamma} \, G_L(r, x; R, x') J_0(R, x') + \phi_h(r, x) \tag{3.2}$$

A key difference[16] here from the Euclidean scenario (other than in the explicit form of the Green's function) is the presence of the homogeneous solution $\phi_h$. The homogeneous solution is annihilated by the wave operator, but unlike in Euclidean signature where such solutions are divergent at infinity, now it is regular everywhere including at infinity (where it vanishes). This means that the general solution with source can include such a piece as well. The Green function $G_L$ is the analytic continuation of the Euclidean Green function, and is the bulk-to-bulk Feynman propagator.

- The homogeneous mode $\phi_h(r, x)$ (or equivalent data, namely its value $\phi(R, x)$ at $r = R$) is dual to a state (denoted $|\phi_h\rangle$) in the dual theory. Plugging the solution (3.2) into the on-shell action (2.6), we find

$$S = \int_{\partial \mathcal{M}} d^d x d^d x' \sqrt{\gamma(R, x)} G(R, x; R, x') J_0(R, x) J_0(R, x') +$$

$$+ \int_{\partial \mathcal{M}} d^d x \sqrt{\gamma(R, x)} \phi_h(R, x) J_0(R, x) \tag{3.3}$$

Taking functional derivatives and then setting the source to zero, we get the 1 and 2

---

[16] A secondary difference is that the radial direction $r$ here labels a timelike foliation.

point functions as

$$\langle O(R,x)\rangle \quad = \quad \phi_h(R,x) \tag{3.4}$$

$$\langle O(R,x)O(R,x')\rangle \quad = \quad G(R,x;R,x') \tag{3.5}$$

For the flat space case, these bulk Green's functions are easily evaluated, and we will quote them letter. Note that (3.4) is again precisely the extrapolate dictionary for the condensate whose analogue in AdS is often quoted and is identical except for the radial scaling factor (see, (3.6)).

To summarize, we can make a statement parallel to the AdS picture developed in [5]: in the Lorentzian case the semi-classical partition function (or on-shell action) with the source $J_0(R,x)$ and homogeneous mode $\phi_h(R,x)$ turned on, is equal to the expectation value of the dual theory deformed by the source term, in the state $|\phi_h\rangle$, ie., $\langle \phi_h|e^{i\int_{\partial M} J_0(R,x)O(x)}|\phi_h\rangle$.

**AdS:** Let us briefly compare this with AdS. In Lorentzian AdS, on top of the non-normalizable mode as in the Euclidean case, we also have normalizable modes that are regular in the interior [5]. At the boundary, they behave like

$$\phi_n(z,x) \to z^\Delta \phi_n(x) + \cdots \tag{3.6}$$

These are precise analogs of our homogeneous modes. In particular, the general solution in Lorentzian AdS is of the form

$$\phi(z,x) = \int d^d x' G_{LAdS}(z,x;x')\phi_0(x') + \phi_n(z,x) \tag{3.7}$$

where $G_{LAdS}$ is the Lorentzian AdS bulk-to-boundary Green function. An equation of a similar form can be found in eg., [5]. The parallel with (3.2) is evident.

## 4  Et Cetera

Let us build on these ideas to outline some general features of semi-classical holography. The comments in this section are fairly telegraphic, a systematic presentation and discussion of some physics will be given in the companion paper [14]. The only point we wish to make here is that the formalism allows explicit calculations.

### 4.1  Bulk Reconstruction

When the non-normalizable mode is turned off, in Lorentzian AdS, we have a direct map between the bulk field $\phi(z,x)$ and the boundary state dual to $\phi_n(x)$. Because of the CFT

state-operator correspondence, this means that in AdS this can also be viewed as a map between bulk field and boundary operator. The question of bulk reconstruction is the question of reconstructing a bulk field given this boundary operator [17]. To do this, we use an object called the HKLL smearing kernel [8]. We will be able to define a similar object in our more general context as well.

But before we describe this, we will briefly note a conceptual issue. In a CFT, we have the canonical state-operator correspondence, which gives a map between local operators and states. Here on the other hand, we do not even know that the holographic theory is local, let alone that it is a CFT. But nonetheless, we expect that the theory has local operators (the $O(x)$ in our notation) because we are able to define them via holography[18]. We further know that homogeneous bulk fields $\phi_h(r, x)$ lead to boundary fields $\phi_h(R, x)$. As far as the structure of PDEs go, this data is precisely enough to write down an HKLL kernel with which we can do bulk reconstruction in analogy with AdS. For these reasons, we find it plausible that there exists a class of boundary operators that are naturally associated to the fluctuations of a given semi-classical bulk background [19]. These operators are what we will use to reconstruct bulk fields.

With this understanding, it is straightforward to adapt the approach of [8] to find our smearing kernels. We do this by using a version of Green's theorem to express the bulk field

$$
\begin{aligned}
\phi(r, x) &= \lim_{r' \to R} \int_{\partial M} d^d x' \sqrt{\gamma(r', x')} \Big( \phi_h(r', x') \partial_{r'} G_S(r, x; r', x') - G_S(r, x; r', x') \partial_{r'} \phi_h(r', x') \Big) \\
&\equiv \int_{\partial M} d^d x' \sqrt{\gamma(R, x')} \, K(r, x; R, x') \phi_h(R, x').
\end{aligned}
\tag{4.1}
$$

Here in the first line, $G_S$ is the *spacelike* bulk-to-bulk Green function[20]. The first line is an identity, a version of Green's theorem. The smearing kernel makes it appearance in the

---

[17]It is important to note that the reconstructed operator is to be understood as acting within correlation functions, so reconstruction at the level of operators should be taken with a grain of salt.

[18]Of course their correlators need not be that of a local quantum field theory, which is indeed what is generally expected for the hologram of flat space. Let us emphasize that a non-local theory can have local operators.

[19]Note that if this were not so, it would be puzzling why the bulk theory has a structure isomorphic to the one found in AdS.

[20]The recipe for obtaining the spacelike Green function (when it exists) is as follows. First we write down the most general Euclidean Green function. This object contains two independent constants because our differential equation is second order. We fix the first constant by demanding the correct behavior near the delta-function source. In the conventional Euclidean Green function, the second constant is fixed by demanding that the Green function vanishes at spatial infinity. If we analytically continue this object, we get the Feynman Green function as we alluded to previously. To obtain the spacelike Green's function, we do not demand vanishing at Euclidean infinity. Instead we analytically continue, and then demand that the Green function vanishes at non-spacelike separation, which fixes the second constant and uniquely fixes the Green function as the spacelike Green function. This approach was developed in [8, 11].

second line, where we read it off based on our knowledge of the first line, and $\phi_h(R, x')$. For a second order PDE, $\phi_h(r', x')$ and $\partial_{r'}\phi_h(r', x')$ are generically independent data at $r'$, but this is where the fact that we are working with homogeneous modes plays a key role. This knowledge enables us to express the latter in terms of the former, which is necessary for the definition in the last line to make sense.

The kernels constructed this way can have subtleties in coordinate space due to divergences (see eg., the discussion in section 4 of [8]) and non-uniqueness, which lead one to interpret them as distributions (see discussion and references in eg., [12]). In practise, this means that suitably defined reconstruction kernels of the form $\mathcal{K}(r, x; R, \ell)$ are often better defined as functions, where $\ell$ stands for the Fourier space of the boundary coordinates $x'$, see eg., [13] for an AdS discussion.

Let us also note that the mode expansion of the homogeneous solution gives us an alternative approach to constructing a bulk reconstruction kernel. We will illustrate this in the next section for flat space. It will be interesting to see if these two constructions of the smearing kernel yield identical objects in flat space [14][21].

## 4.2   Adding Interactions

Adding bulk interactions and perturbatively determining higher point functions in Euclidean space follows an isomorphic picture to what happens in AdS. Witten diagrams are adapted trivially. Witten diagrams involve bulk-to-bulk and bulk-to-boundary correlators in AdS: the former remain unchanged here, and the latter follow our discussion in a previous section relating our source/correlator prescription to the conventional AdS source/correlator prescription. Higher point boundary correlators follow. We will present some of the details in [14].

In the Lorentzian case, a similar statement can be made for bulk reconstruction as well. When the bulk theory has interactions, one can modify the bulk reconstruction procedure parallel to what is done in AdS. A key equation as far as we are concerned is, say eqn. (3.14) (see also eg., eqn. (3.15)) in [12]. Note that this equation is an identity and holds equally well in our case also. It follows immediately that we can correct the bulk reconstruction procedure order by order in the bulk coupling, in a systematic way.

These remarks demonstrate that it is straightforward to add perturbative bulk interactions in computations of correlators as well as in bulk reconstruction in our approach.

---

[21]This is fairly easy to verify in AdS, because asymptotically the radial dependence factorizes.

## 4.3  Example: Flat Space with $\mathbb{R} \times S^2$ Screen

The above discussion has largely been quite general, so let us present a concrete example where explicit calculations are possible. A simple and potentially highly interesting example is the case of Minkowski space, $M_{3+1}$. See (3.1) for the Lorentzian version of the metric, the Euclidean version has positive sign for the time part. The screen is at radius $r = R$. In this letter, we will present a few simple results, details and more complete results can be found in the companion paper [14].

Because of the homogeneity of flat space, the Euclidean, Lorentzian and Spacelike massive scalar Green functions can be written in terms of geodesic lengths [8, 11]

$$G_E(\sqrt{\sigma}) = \frac{m}{(2\pi)^2} \frac{K_1(m\sqrt{\sigma})}{\sqrt{\sigma}}, \quad G_S(\sqrt{\sigma}) = \frac{m}{(2\pi)^2} \frac{\pi}{2} \frac{I_1(m\sqrt{\sigma})}{\sqrt{\sigma}} \tag{4.2}$$

where $\sigma$ is the (Euclidean/Lorentzian) geodesic length. Explicit expressions for these geodesic lengths in terms of coordinates are straightforward [14]. $K$ and $I$ are the modified Bessel functions of the first and second kind respectively. The Lorentzian (Feynman) Green function is related to the Euclidean Green function via $G_L(\sqrt{\sigma}) = iG_E(\sqrt{\sigma + i\epsilon})$. When treating these as bulk-to-boundary propagators, we treat one of the terminal points of the geodesic segment to be at the screen $(R, x')$. Using these it is trivial to write explicit expressions for the holographic correlators immediately from (2.7) and (3.5) using the $G_E$ and $G_L$ above.

Using the mode expansion of the homogeneous solution

$$\phi_h(r, t, \Omega) = \sum_{l,n} \int_{\omega > m} \frac{d\omega}{2\pi} e^{-i\omega t} a_{l,n}(\omega) \frac{J_\nu(r\sqrt{\omega^2 - m^2})}{\sqrt{r}} Y_{l,n}(\Omega) + h.c., \tag{4.3}$$

an HKLL-like smearing function in momentum space is easy to read off as

$$\tilde{K}(r, t, \Omega; R, \omega, l, n) = \frac{1}{R^2} \sqrt{\frac{R}{r}} \frac{J_\nu(r\sqrt{\omega^2 - m^2})}{J_\nu(R\sqrt{\omega^2 - m^2})} e^{-i\omega t} Y_{l,n}(\Omega) \tag{4.4}$$

where $\nu = \frac{1}{2} + l$. The reconstruction works via

$$\phi_h(r, t, \Omega) = \sum_{l,n} \int_{\omega > m} \frac{d\omega}{2\pi} K(r, t, \Omega; R, \omega, l, n) \tilde{\phi}_h(R, \omega, l, n) + h.c. \tag{4.5}$$

where $\tilde{\phi}_h(R, \omega, l, n) \equiv a_{l,n}(\omega) \frac{J_\nu(R\sqrt{\omega^2 - m^2})}{\sqrt{R}}$. The pre-factor $\frac{1}{R^2}$ in (4.4) arises because $\sqrt{\gamma(R, x')} = R^2 \times \sin\theta$.

## 5  Comments and Future Directions

We presented a general prescription for defining semi-classical holography in a fairly large class of spacetime regions.

In Euclidean geometries our discussion was quite general, but is somewhat different from some previous efforts. For example, holographic correlators in Euclidean flat space with the metric $ds_{d+1}^2 = d\rho^2 + \rho^2 d\Omega_d^2$ have been discussed in [16, 17], where a Dirichlet boundary condition was imposed at a boundary cut-off, and the boundary value was taken as the source for the dual theory. Note that this is distinct from our prescription: the key difference[22] at the level of Green functions is that the prescription of [16, 17] uses the Dirichlet Green function (this object depends on the screen, and is defined to vanish there), while ours uses the Euclidean Green function that vanishes at spatial infinity whose analytic continuation leads to the Feynman Green function in the Lorentzian signature.

These two approaches can be viewed as two separate ways to generalize the usual holographic correspondence in AdS, to other geometries. In (Euclidean) AdS, the boundary value of the bulk field and the boundary limit of a bulk source are essentially the same object [23]. This coincidence of two logically distinct objects in AdS means that there are two possible paths to generalize the holographic prescription to other geometries. The usual philosophy, whose concrete realization can be found in [16, 17], adopts the stance that the boundary value of the bulk field is still the object that should be viewed as the source for the dual theory, even when we are not in AdS. We have instead taken the perspective that the bulk source localized at the boundary/screen is what should be interpreted as the dual source. These two perspectives are largely indistinguishable in AdS because the $z$-dependence of the solution factorizes out near the AdS boundary. But away from AdS, they are distinct.

Our prescription has a few features we find attractive:

- The Euclidean Green function we work with is not defined via a screen-dependent condition, it is a property of the theory. The Dirichlet Green function on the other hand is defined via a vanishing condition at $r = R$. Note that at the conformal boundary of AdS this issue is invisible because it is at infinity.

- Our Green function when Wick rotated to Lorentzian signature, ends up being the Feynman Green function, which is what we would like as the standard Lorentzian bulk propagator. The spacelike Green function that we use for bulk reconstruction is also very closely related to the Euclidean Green function, as we explained earlier. The

---

[22]Note also the secondary difference that when you use a spherical (as opposed to cylindrical) holographic screen, there is no natural time coordinate one can Wick rotate.

[23]As we have been careful to emphasize, they are not *exactly* the same object: the boundary value of the bulk field is the dual source, *after* multiplication by $z^{\Delta-d}$. A closely related fact is that the bulk-to-boundary Green function in AdS is not quite the bulk-to-bulk propagator with one point at the boundary, but its derivative in the holographic direction (see eg., [14]). This is expected in a Dirichlet Green function [16].

analytic continuation of the Dirichlet Green function on the other hand, does not lead to a causal propagator.

- In Lorentzian signature, the homogeneous mode that we find is a very natural analogue of the normalizable mode in AdS. There is no natural analogue in the Dirichlet prescription.

- The homogeneous mode allows us to do HKLL-like bulk reconstruction, entirely parallel to Lorentzian AdS.

- The extrapolate and differential dictionaries match in our prescription. As we discussed in the main text, this is not true for the Dirichlet prescription.

- More philosophically, holographic duality implies that there is only one underlying theory, so it is perhaps natural that the source for the boundary theory also has an interpretation as a source for the bulk theory.

There are many directions here worth developing, some of which will be presented elsewhere. Our observations suggest that it is worth re-thinking mechanics and field theory in bounded regions along the lines suggested in this paper. In particular, instead of boundary terms (for well-defined variational principles etc.), it might be instructive to consider an auxiliary system with sources localized at the screen in an otherwise unbounded space(time). In other words, instead of the standard particle mechanics problem with "Dirichlet" boundary conditions at the end points,

$$S_D^p = \int_{t_1}^{t_2} dt \left( \frac{1}{2} \dot{q}^2 - V(q) \right), \tag{5.1}$$

it may be interesting to consider something like

$$S^p[J(t_1), J(t_2)] = \int_{-\infty}^{+\infty} dt \left( \frac{1}{2} \dot{q}^2 - V(q) \right) + J(t_2)q(t_2) - J(t_1)q(t_1). \tag{5.2}$$

Natural generalizations of this to field theory and gravity, clearly exist. Various boundary related themes of a similar flavor have recently been investigated in the context of holography [18]. A closely related question is whether these considerations can be further adapted to say something useful about cosmological backgrounds.

Let us close by presenting a highly incomplete list of open questions/thoughts, some more accessible than others:

- We have only investigated scalar fields in this paper. But clearly, similar constructions must exist for gauge fields and gravitons as well (at least at the linearized level), both when it comes to holographic correlator calculations, as well as for bulk reconstruction [19].

- Continuing in a similar line – for the fully non-linear metric, localizing a source on a codimension-1 surface is in many ways an inverse problem to the familiar Israel junction conditions [22]. In the latter, what one does is to try to match spacetimes across a codimension-1 surface, only to find that one needs to place a singular stress tensor at the interface. What we are interested in is the inverse problem, namely to figure out how the full spacetime responds when a source is placed on the codimension-1 surface. Note that this is a significantly harder problem than the scalar problem we considered in this paper, because Einstein equations are highly non-linear. But especially in situations with symmetries, progress can be made [23].

- Rather remarkably, it turns out that codimension-1 sources are special in general relativity: only they allow the possibility of being interpreted as solutions of Einstein's equations (at least in the distributional sense). Higher codimension sources cannot be accommodated this way; this was proved by Geroch and Traschen [24]. From the perspective of our work, this has the eminently natural interpretation that codimension-1 sources are where holography is realized.

- The boundary value of the bulk field is a source for the dual theory in AdS/CFT. But we saw for the AdS scalar field that its *bulk* codimension-1 source, when near the AdS boundary is essentially *also* the boundary value (2.13). It will be very interesting to see if a similar statement holds true also for the bulk metric in some suitable sense, in AdS. At least at the linearized level, where we treat the on-screen source as a perturbation, we expect that one can make progress on this problem.

- It will be interesting to understand the relevance of the Legendre transform of the action/partition function we have considered here. See [7] for related discussions in AdS.

- Can we relate our flat space on-screen holographic correlators to S-matrix elements in flat space [20]? Our discussion was in coordinate space. What about correlators in momentum space or Mellin space? The on-screen correlators for flat space that we have defined should behave like S-matrix elements when the energy scale $E$ is such that $E \times R \gg 1$, in precisely the same way that CFT correlators behave like S-matrix elements when $E \times l \gg 1$ where $l$ is the AdS scale [21].

- What is the significance of the state-operator correspondence for holography? We feel that this question has not been investigated with enough gravitas.

- AdS/CFT emerged in the decoupling limit of the brane and the bulk, and this is related to the conformal invariance of the duality. In our general holographic setting,

there is no decoupling and there is no scale independence. We do not expect the on-screen-correlators to be those of local theory, certainly not at finite cut-off. It may be interesting to introduce a weak dependence on $r$ in our sources. When the dependence becomes delta-function localized on $r = R$ it will reduce to the discussion in this paper.

- In many discussions of holography away from AdS (say, in flat space), the discussion is often based on symmetries like BMS. This can be viewed as a kinematics-first approach, the dynamics is often secondary. Note that our approach is in this sense, complementary. We are directly dealing with correlators on the screen, and therefore explicitly dealing with (perturbative in the bulk coupling) dynamics.

- In a theory with dynamical gravity, fixing the location of a codimension-1 screen in some coordinates might seem fishy. This would indeed be a cause for concern if our approach claimed to be fully non-perturbative. But our description is (at least at this stage) best viewed as a description valid to all orders in perturbation theory in the bulk. To all orders in perturbation theory, we expect the bulk metric to be a well-defied object[24]. Given a metric and a coordinate system for it, a surface defined in that coordinate system has intrinsic meaning: the coordinate description will change as we do coordinate transformations, but the surface as a geometric object remains fixed. See [25] for a recent discussion on surfaces in the bulk of quantum gravity.

- A related comment is that our results have a certain robustness to them: our prescription is based largely only on the (fairly rigid) structure of PDEs and their solutions in classes of spacetimes.

- We have not discussed black holes at all, but it seems likely that many of the statements about black holes in AdS/CFT can be adapted here. It will be instructive to clarify the precise sense in which there are differences.

- We have also not discussed cosmology, but it is clear that correlators on spatial slices built analogously to what we have done in this paper for timelike slices, may be useful for dealing with cosmological holography. This maybe related to the dS/CFT correspondence [26]. Note however that the extrapolate and differential dictionaries do not [15] match in dS/CFT.

# 6 Acknowledgments

We thank Glenn Barnich, Oleg Evnin, Feroz Hatha, Dileep Jatkar, Ajay Mohan, Vyshnav Mohan, Onkar Parrikar, Charles Rabideau, Thomas Van Riet and especially Ashoke Sen for

---

[24]Note that it need not satisfy the Einstein equations, of course.

comments/discussions. CK thanks the audiences at KU Leuven and HRI Allahabad for feedback on talks based on this material.

## A    Alternate Prescription

In the main text, we gave a prescription by integrating the on-shell action all the way to infinity in the bulk. We believe this is more natural (largely because this results in a match between extrapolate and differential dictionaries), but it may also be of some interest to consider a prescription where we integrate only up to the screen. In this appendix, we present the prescription and some results for that.

In the Euclidean setting, we should compute the bulk on-shell (semi-classical) partition function, or (in practice) the classical action, of this solution *within the holographic screen*. It is a straightforward fact that for a scalar two-derivative theory, this will be of the form

$$S_{bulk} = \lim_{r \to R} \int_{\partial M} d^d x \, \sqrt{\gamma(r,x)} \, \phi(r,x) \partial_r \phi(r,x) \tag{A.1}$$

and using (2.3) this can immediately be written as a functional of $J_0(R,x)$. This lets us calculate correlation functions of operators $O(x)$ dual to the sources $J_0(R,x)$ via functional derivatives with respect to the $J_0(R,x)$, in a manner very similar to AdS. When evaluated at zero-source, this procedure leads to a vanishing 1-point function, and a two-point function of the form

$$\langle O(x')O(x'')\rangle = \lim_{r \to R} \int_{\partial M} d^d x \sqrt{\gamma(r,x)} \, \partial_r \{ G_E(r,x;R,x'') \, G_E(r,x;R,x') \}. \tag{A.2}$$

Even though our notation does not emphasize it, it should be kept in mind that the dual operators $O$ and their correlators can depend on $R$.

In the Lorentzian setting, the homogeneous mode $\phi_h(r,x)$ is dual to a state (denoted $|\phi_h\rangle$) in the dual theory. Taking the functional derivative of the action with respect to the source gives us

$$\langle \phi_h | O(y) | \phi_h \rangle_{J_0=0} = \lim_{r \to R} \int_{\partial M} d^d x \sqrt{\gamma} \, \partial_r \{ \phi_h(r,x) \, G_L(r,x;R,y) \} \tag{A.3}$$

$$\langle \phi_h | O(y) O(z) | \phi_h \rangle_{J_0=0} = \lim_{r \to R} \int_{\partial M} d^d x \sqrt{\gamma} \, \partial_r \{ G_L(r,x;R,y) \, G_L(r,x;R,z) \} \tag{A.4}$$

Note again that the presence of the homogeneous piece leads to a non-trivial 1-point function even when the source is zero. Again, we have suppressed the $R$-dependence of the left hand side.

With the modified prescription, the Euclidean boundary correlator for Minkowski space (cf., section 4.3) is more complicated and takes some work to calculate:

$$\langle O(t', \Omega', R) O(t'', \Omega'', R) \rangle = \tag{A.5}$$

$$= -\frac{m^2}{4\pi^4} \int dt d\Omega \; mR^3 \left( \Xi' \frac{K_1(m\sqrt{\Lambda''}) \, K_2(m\sqrt{\Lambda'})}{\Lambda'\sqrt{\Lambda''}} + (\Lambda'', \Xi'' \leftrightarrow \Lambda', \Xi') \right)$$

To specify the notation, let us note that $d\Omega$ contains the $\sin\theta$. We also have

$$\Lambda'' = 2R^2 + (t - t'')^2 - 2R^2(\cos\theta \cos\theta'' + \sin\theta \sin\theta'' \cos(\phi - \phi''))$$

$$\Lambda' = 2R^2 + (t - t')^2 - 2R^2(\cos\theta \cos\theta' + \sin\theta \sin\theta' \cos(\phi - \phi'))$$

$$\Xi' = -1 + \cos\theta \cos\theta' + \sin\theta \sin\theta' \cos(\phi - \phi')$$

$$\Xi'' = -1 + \cos\theta \cos\theta'' + \sin\theta \sin\theta'' \cos(\phi - \phi'').$$

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
