# Peer review of "A General Prescription for Semi-Classical Holography"

_SciPost Physics_

## Round 2 · Referee Report · Anonymous · 2020-6-9

Strengths

1. A very interesting and relevant study with quite a lot of possible applications.
2. The claims are clearly stated.
3. Sections 1, 2 and 3 are very nicely written

Weaknesses

1. Usefulness of the bulk-screen duality for AdS is not sufficiently stressed with examples.

2. The claim that the proposal can be used for more general class of spacetime, including the flat space is not established in full detail.

3. Related to the previous point, section 4.3 is extremely brief to convey the achievement the authors claimed at the beginning of this paper.

Report

First of all, I would like to thank the authors for coming up with an issue that is very much relevant for a large class of holographic interests.

The paper presents a proposal for a bulk-screen dictionary when a source is present on a holographic screen at a finite radial distance in the bulk, instead of being on the boundary as is the case for usual AdS/CFT. They further provide with a HKLL-like bulk reconstruction formula using the homogeneous mode obtained through Lorentzian continuation which can be interpreted as the normalizable mode in the language of traditional holography. Finally based on the fact that the programme can be carried out without any reference to the physical boundary of the spacetime, the authors claim that the programme can also be useful to establish such a ''holographic dictionary" for a larger class of spacetime including one in flat spacetime.

First of all, such a formulation of holography in terms of source localized on an arbitrary holographic screen is useful even in AdS. The holographic screen divides the spacetime into two parts and therefore becomes a natural candidate to understand so called braneworld models (eg. Ranndall-Sundrum model) when the brane plays the role of a localized source in the bulk. In fact it would be nice to compare with some related work in literature, namely one in hep-th/0002091. The computation of the on-screen correlator presented in this paper is very similar in spirit to that treatment discussed in the present manuscript. A comparison and applicability of the current proposal in such scenarios will enhance the usefulness of the same. There are also some recent work including 2001.07433 where a computation of a braneworld propagator was presented where in two sides of the brane have two different different AdS's with different cosmological constants. I believe the present proposal would be extremely useful even in those situations. The difference would perhaps be in this case one would need a non-trivial junction condition. I would urge the authors to comment on the possibility of such generalization. All in all, the proposed bulk-screen duality for AdS needs some examples to establish its usefulness. I found some brief comments on this in one of the bulleted points in section 5, however, a detailed account of that would be useful comparing with the existing literature mentioned above.

The same comment applies regarding the claim that the proposal can be used for more general class of spacetime, including the flat space. The flat space example presented in section 4.3 is very brief. Although the authors refer to an upcoming work, this is insufficient as evidence towards the afore-mentioned claim (which is one of the most promising claims made at the beginning).

Third, the authors do not make any comments on the holographic renormalization. For AdS, when the holographic screen is pushed towards the boundary it becomes necessary. A comment on this issue is necessary to my opinion. In particular it would be interesting to understand the interpretation of "homogeneous modes" for graviton fluctuations.

Requested changes

1. Comments on the usefulness of the proposal to compute correlators with sources on a holographic screen in AdS. Comparison with similar previous attempts in the context of braneworld models, in particular.

2. Major revision of section 4 is needed. The claim regarding the applicability of the proposal flat space holography needs to be discussed in detail evidence and possible applications.

3. Comments on holographic renormalization.

---

## Editorial Decision

awaiting_resubmission